# Diversity of Phosphate Chemical Forms in Soils and Their Contributions on Soil Microbial Community Structure Changes

**DOI:** 10.3390/microorganisms10030609

**Published:** 2022-03-13

**Authors:** Amandine Ducousso-Détrez, Joël Fontaine, Anissa Lounès-Hadj Sahraoui, Mohamed Hijri

**Affiliations:** 1Unité de Chimie Environnementale et Interactions sur le Vivant (UCEIV), Université du Littoral Côte d’Opale, UR4492, SFR Condorcet FR CNRS 3417, 62228 Calais, France; amandine.ducousso@umontreal.ca (A.D.-D.); joel.fontaine@univ-littoral.fr (J.F.); anissa.lounes@univ-littoral.fr (A.L.-H.S.); 2Département de Sciences Biologiques, Institut de Recherche en Biologie Végétale (IRBV), Université de Montréal, Montréal, QC H1X 2B2, Canada; 3African Genome Center, Mohammed VI Polytechnic University (UM6P), Ben Guerir 43150, Morocco

**Keywords:** phosphorus, chemical forms, inoculant engineering, microbial community, plant biostimulants

## Abstract

In many soils, the bioavailability of Phosphorus (P), an essential macronutrient is a limiting factor for crop production. Among the mechanisms developed to facilitate the absorption of phosphorus, the plant, as a holobiont, can rely on its rhizospheric microbial partners. Therefore, microbial P-solubilizing inoculants are proposed to improve soil P fertility in agriculture. However, a better understanding of the interactions of the soil-plant-microorganism continuum with the phosphorus cycle is needed to propose efficient inoculants. Before proposing further methods of research, we carried out a critical review of the literature in two parts. First, we focused on the diversity of P-chemical forms. After a review of P forms in soils, we describe multiple factors that shape these forms in soil and their turnover. Second, we provide an analysis of P as a driver of microbial community diversity in soil. Even if no rule enabling to explain the changes in the composition of microbial communities according to phosphorus has been shown, this element has been perfectly targeted as linked to the presence/absence and/or abundance of particular bacterial taxa. In conclusion, we point out the need to link soil phosphorus chemistry with soil microbiology in order to understand the variations in the composition of microbial communities as a function of P bioavailability. This knowledge will make it possible to propose advanced microbial-based inoculant engineering for the improvement of bioavailable P for plants in sustainable agriculture.

## 1. Introduction

Global demand for agricultural crops continues to increase and global food production is largely dependent on intensive agricultural management. In conventional agriculture, high-yielding crop varieties, irrigation, pesticides, and fertilizers are frequently used as common farming practices to attain higher crop yields. For instance, organic and/or inorganic fertilizers are used to provide available Phosphorus (P) compounds to plants by supplementing phosphate ions and to rapidly satisfy the nutrient requirements of crops. However, supporting plant growth and yield optimization has raised an overuse of P fertilizers, increased production costs and exerted negative impacts on agricultural soil quality and surrounding environments; in particular, surface water eutrophication [1]. In addition to these negative environmental impacts, an important proportion of P inputs is rapidly sequestered by P fixation, or precipitation by the soil matrix into insoluble complexes, inside which P is no longer directly available for biological assimilation by crops, further reducing fertilizer efficiencies.

Consequently, efforts have been made to develop alternate strategies to reach adequate levels of bioavailable P in soils and improve P uptake by plants. Among them, crop-breeding programs incorporating the selection of beneficial plant-microorganism interactions to breed ‘an optimized crop-holobiont’ are promising alternatives. Agro-inputs exploiting soil microbiota ecosystem services inside microbial inoculants are also reported as a natural solution for promoting P cycling and P bioavailability [2]. Indeed, the soil microbial biomass is considered as a temporal sink and source of P compounds [3]. In addition, the soil microbial functions catalyze key processes in soil P turn-over and cycle, from various soil organic and inorganic pools [4]. In particular, some microorganisms, referred to as plant growth promoting microbes (PGPM), are identified as stimulating plant growth via direct and/or indirect mechanisms [5,6,7] and have been proposed as agriculturally beneficial microorganisms.

Among PGPM, phosphate solubilizing microbes (PSM) are identified as microbes exhibiting in vitro the ability to solubilize insoluble inorganic P-compounds producing available P-forms [8,9,10,11,12,13]. Thus, screening microbes for P-solubilizing taxa from field soils (P-deficient soils, or P-rich soils in mining zones) for use as microbial-based fertilizers in agronomic practices is recognized as an area of interest and has gained worldwide interest in past decades [2,14,15,16,17,18]. Some researchers even suggest the inoculation of phosphate-solubilizing bacteria (PSB) in soils can reduce the phosphate fertilizer application rate by 50% [19,20]. Different formulations have been proposed: engineered with a single strain or as a consortium of various microbial strains exhibiting diverse plant growth promoting properties (including P-solubilization). Nevertheless, while there is evidence of the efficacy of microbial formulations for soil fertility and plant productivity in different cropping systems, their performance as bio-fertilizers in field experiments remains debated by some authors who call for further evaluation of the PSM concept [21]. In addition, the impact of microbial inoculants on native communities in field applications is sometimes interrogated [22].

To progress in formulating widely efficient microbial-based inoculum, a deeper understanding of how P and the bacterial and fungal multi-kingdom community interact is required. In this context, we first review recent data concerning the drivers of P speciation in soils and of the dynamics of P-forms. Second, we report a growing set of recent studies investigating the contribution of P in shaping the responses of soil microbial communities, with a focus on fungal and bacterial microbiomes. Finally, we question how research in P and soil microbiome interactions is currently constrained from recent advances in our identification and understanding of soil P species diversity.

## 2. Diversity of Phosphorus-Forms and P-Dynamic in Soils

### 2.1. Back to Orthophosphate Ions and Distribution of P in Different Chemical Species

Phosphorus is not observed naturally as a free elemental form; it is highly reactive, combining rapidly with other elements. Notably, simply exposing it to air will stimulate a chemical reaction with oxygen. Therefore, P has many degrees of oxidation, from −III to +V [23]. In natural systems, P is almost exclusively present in the (+V) oxidation, i.e., as H_3_PO_4_, the tri-protic orthophosphoric acid, referred as phosphoric acid. It is a very polar molecule which makes it highly soluble in water. The dissociation successively produces dihydrogen phosphate anions (H_2_PO_4_^−^), hydrogen phosphate anions (HPO_4_^2−^) and the tetrahedral oxy-anion orthophosphate (PO_4_^3−^) which are the fully dissociated orthophosphate anions of H_3_PO_4_ [23]. However, HPO_4_^2−^ and H_2_PO_4_^3−^ ions (referred to as iP) are the major mobile P forms in soil solution [24], in proportions relative to the soil pH, and linked to the dissociation constants of the successive protonization/deprotonization reactions of orthophosphoric acid [23,25]. For instance, when the soil pH is less than 7.0, H_2_PO_4_^−^ is the predominant form in the soil. Moreover, the HPO_4_^2−^ and H_2_PO_4_^3−^ anions are the dominant bioavailable P forms, i.e., the fraction of total P in soil that is readily available for acquisition by microbes and plant roots [26]. However, iP is a highly reactive ionic form that interacts with numerous chemical compounds. Thus, much less than 1% of the total P will normally be found dissolved in the soil solution [10,24], and the soluble iP concentration rarely exceeds 10 µM in soil solutions [8,24,27]. The low availability of phosphate, due to the poor solubility and mobility of soil P, frequently impairs plant growth and associated-metabolic pathways, and is therefore a major factor constraining plant performance in many natural and agricultural soils worldwide.

In addition, iP can be incorporated into a wide range of soil constituents, inducing a complex P speciation (i.e., P distribution in different chemical species), classically divided into two pools: inorganic P (inorgP), organic P (orgP) [28,29]. The inorganic pool usually accounts for 35% to 70% of total P in soil, mainly depending on soil age [28,30,31]. Apatite is the most common form of P, forming about 95% of the Earth’s crust’s P, but numerous other mineral P-forms are also observed, varying widely in their chemistry and structure. Indeed, primary apatite minerals exhibit a hexagonal crystal structure with long open channels. In its pure form, F^−^, OH^−^ or Cl^−^ occupies sites along these channels to form fluorapatite, hydroxyapatite, or chlorapatite, respectively. In sedimentary rock, the most common form of natural apatite is francolite, a carbonate fluorapatite. However, the fact that the crystal lattice of apatite is “open” offers the possibility of many substitutions. Additionally, phosphorus can be abundant in many secondary phosphorus minerals. Phosphorus can also be present in trace amounts on the surface of particles such as clays, calcium carbonate, ferric oxyhydroxides or in the crystal lattices of other minerals. Ultimately, P occurs naturally in more than 300 different minerals [29].

Organic P forms (orgP) are characterized by the presence of carbon hydrogen bonds [32]. In natural soils, they are derived mainly from biological processes involving the assimilation of orthophosphate and subsequent release in soil. The orgP fraction classically constitutes 30–65% of the total P in most soils [33,34,35], although it may range from as low as 5% (in mineral soils) to as high as 95% in organic soils (>20–30% organic matter) [30,36,37,38]. OrgP consists of a large variety of compounds which are generally classified into four groups: monoester phosphates, diesters phosphates, phosphonates and organic polyphosphate [32,39]. Phosphate monoesters are the predominant form of orgP in soils under aerobic conditions, which occur mainly as inositol phosphates [40]. The abundance of inositol phosphates is highly variable, but frequently accounts for up to 50% to 80% of the total orgP [41], phytic acid being the most common stereoisomer [32,42]. Phosphate diesters include nucleic acids (i.e., DNA and RNA), phospholipids and teichoic acids [30,39,40]; nucleic acids and their derivatives account for 1% to 3% of total Po [30]. Phosphonates are characterized by a carbon-phosphorus bonds (C-PO_3_^2−^), a bond that gives them great chemical stability. Phosphoric acid anhydrides (i.e., organic polyphosphate) include compounds such as ADP and ATP.

Soil P content results also from immobilized P in biomass. Indeed, the microbial biomass can contain up to 26% (more typically c. 5–10%) of the total soil P [3,43]. Depending on seasonal conditions (especially soil moisture) and the turnover time of microbial biomass, the release of P from the microbial biomass can be significant; it therefore represents an important ‘potential pool’ of bioavailable P [44,45,46].

### 2.2. Geophysicochemical Processes Involved in P Cycling and Speciation

Weathering, sorption and precipitation/dissolution are essential geochemical processes that control P dynamics and speciation in soils [23,47]. Sorption does not refer to a specific mechanism but on the contrary, results from a set of abiotic reactions by which one chemical form becomes attached to another, through the pairing of ions with different soil compounds, with more or less strength, up to precipitation and formation of strong covalent bonds [48]. Sorption is the result of a mineral equilibrium, and two main physicochemical processes of sorption can be distinguished: dissolution/precipitation and adsorption/desorption. However, the mechanisms of sorption and subsequent transformation processes are only approximately known [48,49,50]. Moreover, it is sometimes difficult to distinguish accurately precipitation/dissolution and adsorption/desorption processes in soils because of the continuum between P adsorbed by more or less energetic bonds up to precipitated P crystallized.

Precipitation/dissolution are both very slow processes. The release of iP from apatite dissolution is the first key control on P availability in soil via weathering. The precipitation of negatively charged iP with a cation is observed when the concentrations of iP and cation are sufficiently high. It leads to a decrease in the concentration of iP in solution and the formation of a solid in which iP and the cation are present according to a stoichiometry characteristic of the mineral formed. Then, P-forms remain included in the constituents, reducing the possibility of transfer between soils compartments, towards living organisms or groundwater. On the contrary, adsorption is a fast and easily reversible process. It traps P-forms on the solid soil phase through mechanisms on surface sites and leads to the accumulation of ions at the interface between the solid phase and the liquid phase. Soil sorbents include many compounds that have surface charges variable and sorption occurs through physical or chemical processes such as mainly, anion exchange, precipitation of Ca phosphates, ligand exchange to Al and Fe oxides/hydroxides and edges of alumino-silicate minerals [50]. Compounds having hydroxyl groups, i.e., oxides and hydroxides of iron (Fe(OH)_3_, aluminum (Al(OH)_3_ and calcium (CaCO_3_), that compose coatings on soil particles or are precipitated as interlayers in clays are the major factors for P sorption [51,52,53]. In alkaline soils, the clay content, Ca and Mg are the major factors for P sorption. In acid soils, iron and/or aluminum oxy-hydroxides and alumino-silicates dominate [9,37,54]. Po-compounds (more particularly phytic acid) are also frequently adsorbed by mineral clays and form complexes or precipitates with iron and/or aluminum oxy-hydroxides [52,55] in acidic soils. Organic matter such as fulvic acids in calcareous soils and humic acids in acidic soils can also sorb P compounds through various interactions: physical incorporation in organic matter by electrostatic connections of hydrophobic forces, chemical incorporation, direct or indirect adsorption through cationic bridge forming organic P-metal-organic matter complexes [56].

P compounds which differ significantly in their contributions to soil P availability due to their chemical and biological nature [57,58,59], are classified into different fractions: (i) soluble inorganic and organic P in the soil solution; (ii) labile inorgP and orgP, (iii) insoluble P. Labile P is adsorbed P-forms that are in permanent exchange with the dissolved forms through sorption process and move in and out of the soil solution according to pH, temperature, moisture and concentration. The balance between adsorption and desorption therefore governs the equilibrium with P in soil pore-water and the P-buffer capacity of the soil. Insoluble P includes Fe, Al and Mn oxyhydroxides forms, which constitute the main long term storage P-bearing phases in soils, as well as the insoluble organic P in undecomposed plant, animal, and microbial residues within the soil organic matter [24,26,60,61].

### 2.3. Multiple Drivers Shaped the P-Form Diversity in the Soil Matrix

Diverse measures allow partitioning of P-compounds into pools with different chemical properties that characterize “soil P diversity”, which have been well detailed and accurately quantified [26,47,58,62,63,64]. Quantification of the microbial P pool is also sometimes proposed [39,58,65,66,67,68,69,70]. With various protocols and levels of P fractionation, the authors have collectively identified multiple factors acting as drivers of soil P dynamics and speciation: soil properties, climatic variables or soil management practices. Based on the literature, we propose, in Figure 1, a synthetic view of the multiple drivers that shape P form distribution in the soil-plant-microorganism continuum and that could be used as factors for P management.

#### 2.3.1. Soil Properties as Drivers of P-Speciation

The parent material (i.e., the primary geochemical reservoir of P chemical forms) and pedogenesis during geological timescale appear as essential factors to describe soil P dynamic and cycling [59]. This is notably supported in global maps generated by Yang et al. [65] who estimated the spatial distribution of different forms of P and the total P in soils based on global surficial lithology maps, literature reviews about rock P concentration in parent material, quantification of soil P transformation from conceptual models and depletion of total P during soil development. The general agreement between their estimation about total P and measured total P indicates the parent material the weathering of bedrock and P transport through leaching and erosion determines soil P cycle regulation in the long term. In agreement, from a dataset using ^31^P nuclear magnetic resonance, Deiss et al. [71] highlighted how soil inorgP and orgP compounds responded to edaphic variables and soil weathering stages on a wide geographical scale. Ringeval et al. [68] combined several global datasets to analyze labile inorgP and total P of the soil. These authors thus established that one of the main drivers of the spatial variability of the P content of soils, with variations between total P and labile inorgP, originated from the biogeochemical background of the soil corresponding to the P of natural soils during the conversion to agriculture.

Besides soil geochemical characteristics, shifts in plant and microbial community during ecosystem development are also major drivers of pedogenesis. Notably, these drivers control the accumulation of organic matter and modifications in P sources with the occurrence of an increasing range of soil orgP during long term ecosystem development, relevant as a substrate for further P-cycling and turnover by microbes. In particular, as pedogenesis evolves due to both soil weathering stages and ecosystem development, changes in the complexity, as well as the absolute and relative abundances of orgP and inorgP compounds in soil, are observed [39,59,65]. For instance, chronosequence analyzes of natural soils around the world have shown that through long-term ecosystem development, cations and orthophosphate in young soils [72] are directly derived from bedrock weathering. In contrast, older soils, therefore whose parent rock is more altered, can be profoundly modified with, in particular, a reduction or even a possible depletion of P, mainly in the form of apatite; consequently, during the evolution of soil, P can become a limiting element due to its accumulated losses, but also, due to its immobilization in secondary minerals or its complexation in organic compounds. Besides, it has been underlined that highly weathered soils generally have higher C/P ratios, a lower pH, and greater clay concentrations, which may affect the sorption capacity of the soil, and therefore, P speciation [71]. In addition, Hou et al. [73] concluded that pH and organic C concentration regulate competition between plants for P uptake and secondary minerals speciation. On the other hand, the development of organic matter also affects P-mobility by increasing the capacity of the soil to hold P via metal ligands such as Fe, Al and Ca Stutter et al. [58]. Furthermore, it has been shown that acid phosphomonoesterase activity and soil P fractions (such as labile inorgP, intermediately available inorgP, orgP, occluded P and apatite P) differ significantly during the different stages of forest succession: for instance, immediately available inorgP and orgP were prevalent in the late stages; occluded P was higher in the mid-successional forest and acid phosphomonoesterase activity was enhanced in mid- or late-successional forests [74]. Similarly, a marked increase has been observed in the DNA concentrations, long-chain polyphosphate and phospholipids in the organic horizon of old sites, whereas a peak of concentration in inositol hexakisphosphate was recorded in the intermediate stage sites [75]. In accordance, accumulations of pyrophosphate and polyphosphate have been reported as the result of the incorporation and stabilization of these biologically derived compounds in soil organic matter during ecosystem development [76]. In a more general manner, the soil chronosequences showed that the soil orgP speciation is dominated in aerobic conditions, by phosphate monoesters (mainly inositol phosphates) then by phosphate diesters (notably DNA) and phosphonates, irrespective of parent materials, plant cover and climatic conditions, and that most orgP forms increase with age to reach a peak, then decline with time [72].

In the same chronosequence, an analysis of the dynamics of P over the entire ecosystem taking into account the P of plant and microbial biomass and other pools of P in the soil made it possible to highlight the importance of the microbial pool which accounts for 68~78% of the total biomass P in mature soils where major changes in plant and microbial communities have been observed [72,74].

Soil pH is classically considered to be the major variable of soil chemistry due to its profound impact on countless chemical reactions. For instance, as previously described in this paper, pH affects the speciation of orthophosphate ions. Moreover, because the pH of the soil solution determines the number of charges developed on the surface of clays, organic matter and iron oxides, soil aluminum and the forms of orthophosphate ions, pH is a strong factor that affects P speciation, in particular through sorption [48,49,50,54].

As reported by Ziadi et al. [26], soil available P was also affected by soil physical properties such as texture. In particular, Suñer and Galantini [77] reviewed how soil texture affected P availability by affecting pore-size distribution and pore continuity, which in turn controls soil water availability, gas diffusion and the activity of soil biota. They notably concluded orgP contributed significantly to the total P in predominantly sandy soils, whereas in finer textured soils the inorgP pool was the highest.

#### 2.3.2. Influence of Climatic Variables

Numerous studies have tried to assess how climate variables affect soil P cycle, speciation and availability. These have mainly focused on soil/air temperature and precipitation, as well as aridity, or drought [78]. Notably, Hou et al. [73] evaluated how climate patterns affected soil P cycle and availability in global terrestrial ecosystems using a global database of 760 soils, compiled from 96 published studies. They concluded soil P availability, indexed by Hedley labile inorganic P fractions, significantly decreased with increasing mean annual temperature and precipitation. They also suggested that temperature, precipitation and aridity all affected soil P availability, with the effects varying essentially with soil particle size. Deiss et al. [71] established that climatic variables (mean annual precipitations and temperatures) diversely regulated soil inorgP and orgP pools, in combination with edaphic variables primarily determined by the parent material and soil forming factors. Hui et al. [79] noted that orgP concentration decreased with increasing temperatures.

Related to precipitation, the conclusions of Deiss et al. [71] underlined inorgP decreased and orgP should increase as precipitation increases; as the precipitation increased, proportions of orthophosphate decreased and pyrophosphate increased. Liebisch et al. [67] observed that seasonal patterns of fluctuations in available inorgP and microbial P, were mainly driven by soil moisture during a season. In a warm temperate forest, Zhang et al. [73] investigated how drought altered soil P dynamics and bioavailability. In a four-year field drought experiment, these authors found that drought significantly reduced soil calcium phosphate content and that this decrease was correlated with the formation of secondary minerals (Fe/Al oxides) accompanied by an increase in inorgP and orgP. In the analysis by Hou et al. [73], they suggested aridity, mainly determined by mean annual temperature, has a relatively minor effect on soil available P. Moreover, their analysis concluded that climate effects on soil available P contrasted between low-sand soils and high-sand soils, suggesting that temperature and precipitation interact with soil particle size to affect soil P availability. Ziadi et al. [26] highlighted that freeze–thaw cycles can stimulate soil mineralization and could therefore be one factor regulating soil available P concentration in early spring. They also emphasized that increased soil available P during the rewetting phase could result from release of organic P from lysed cells of microbial biomass.

#### 2.3.3. Influence of Land Uses and Soil Management Practices

There is an increasing body of literature concerning how land use and soil management, through various agronomic practices, may drive P dynamics, including affecting diversity and abundance of P species. In this context, various environments are studied; arable lands, grassland, pasture soils, grazing sites, forest and agricultural management such as crop rotation or tillage [39,58,66,68,70,72,78,80]. Interestingly, Stutter et al. [58] estimated that aspects of land use were influential on P species and concentration, after examining thirty-two temperate soils and uses. They also noticed a complex interplay with key soil properties, controlling soil P accumulation and the P species occurring. Nevertheless, despite the large variation between soils, sites and management factors, they concluded that arable soils were dominated by inorganic orthophosphates, with P-monoesters to a lesser extent. Diesters, polyphosphates and microbial P were limited. Orthophosphates and monoesters dominated in the same proportions, in intensive grasslands, while in contrast, P is dominated by orgP, including monoesters, diesters and polyphosphates, as grazing becomes extensive and microbial P increases.

The impact of tillage, which involves plowing and harrowing to mix P imported in amendments and P recycled from crop residues throughout the plow layer, is also the subject of many studies [26,66,81,82]. For example, Xomphoutheb et al. [82] investigated the effects of different tillage systems on phosphorus and its fractions in the rhizosphere and non-rhizosphere. They observed that available P is increased in the non-rhizosphere area under different tillage methods, while the concentration of available P is reduced in the rhizosphere. They concluded that soil management minimizing soil disturbance increased the abundance of different P fractions and improved P availability. In agreement, Oberson et al. [83] reported in their review that total and available P concentrations are higher in the topsoil than in the lower soil horizons with no-tillage; as well, organic P and microbial P was higher in the topsoil of no-tillage soils than in plowed soils.

#### 2.3.4. Interrelations, Coupling and Feedback between the Different Environmental Variables

While multiple drivers of P form distribution have been identified, the data on how these different drivers combine to explain P distribution and availability are still lacking, and research faces several difficulties. Indeed, the environmental drivers (soil properties, climatic variables, anthropogenic uses and agricultural practices) are interconnected variables with feedbacks between them [58,69,71,73,81,84,85]. In consequence, P compounds partitioning responds to different combinations of explicative variables and the relative contribution of a given driver taken individually is extremely difficult to decipher [86]. In addition, data analysis is made difficult because of the diversity of the analytical methods used to identify the P diversity and P fraction numbers. Moreover, the chemical fractionation procedures have been developed to evaluate the main P forms in soil, not the totality of the various P compounds. Moreover, these procedures were based on differences in the chemical resistance and biological susceptibility to degradation, giving rise to descriptive models, which in some cases, may exceed field validation possibilities. Together, such facts make data analysis about P speciation more complex.

## 3. Phosphorus as a Driver of Shifts in Soil Microbiomes

### 3.1. A Large Diversity of Experimental Designs Has Been Used to Track P as a Driver of Microbial Community Assemblages

From the literature, Table 1 compiles selected works that described P-dependent microbial community profiling, where analyses of how P impacts microbial communities have been addressed in different ways. For example, investigations studied different biotopes in distinct plant species using various experimental designs, including different agricultural practices, with or without repeated P inputs over a more or less long period. In addition, different analytical tools are used to describe microbial communities.

For instance, the impact of P inputs on soil microbial communities has been investigated in different global terrestrial ecosystems, including grasslands [87], arctic tundra [88], pastoral agricultural systems [89], forests [90], or estuarine areas [91], and in greenhouse experiments [92]. Authors have variously analyzed the impact of P within the bulk soil, and the different plant compartments (rhizosphere, roots, lateral versus axial roots), highlighting how P diversely affects microbial communities in these different ecological niches [92,93,94,95]. Across these studies, various plant species were included, and the evidence is well documented that the microbial response to P inputs is strongly influenced by the host plants. For example, it appears that non-mycorrhizal *Arabidopsis thaliana* and mycorrhizal *Petunia* sp. formed different microbial associations under the same low-P conditions [96]: notably, Burkholderiales (ex: *Candidatus Accumulibacter*), Rhodocyclales (ex: *Dechloromonas* sp.) and Bdellovibrionales are the most abundant *Arabidopsis* root bacteria while the *Petunia* root bacteria only included Burkholderiales (ex: *Candidatus Glomeribacter gigasporarum*) and Rhodocyclales as prevalent members. Similarly, addition of P significantly altered the microbial community composition in an old-growth tropical forest, but the same P treatment had no effect in a pine forest [90].

Changes in the microbial community composition have been examined with different types of P fertilizers: mineral fertilization with superphosphate (SP) or triple superphosphate (TSP) [89,92,97,98,99], manure fertilization [100,101], combinations of mineral and organic fertilizers, rock phosphate [89,98,102,103], or P-nutritive solution [90,96,104]. Notably, Gumiere et al. [103] have suggested that P sources can explain 39.1% and 45.77% of the variability in the bacteria and fungi community assemblages, respectively.

The impact of P levels on the microbial community has also been studied. High P levels (from the increase to excess) or low P levels (in nutrient limiting conditions, P-deficient soils or in depletion conditions) are both identified as inducing shifts in soil microbial communities [4,89,97,104,105]. Interestingly, taxa whose abundances varied substantially in response to P level or P fertility gradient were identified in both climate chambers with nutritive solutions [96], as well as in fields with contrasting histories of P fertilizer treatment [86].

In addition, there are experiments designed to target the impact of long-term P fertilization (i.e., multiple application patterns over a long period of time) under different P supplementation histories [92,95,98,100], or, instead, short-term fertilization experiments [93,94,95,102,103,105]. In both designs, shifts in soil bacterial and fungal communities were described. For example, as part of a proteogenomic comparison of soil microbial communities, Yao et al. [42] reported the adaptation of soil communities to different P levels as a function of phosphorus fertilization. This adaptation was reflected in particular by changes in community structure, regulation of enzymatic abundances and the gain or loss of metabolic capacities.

Finally, determining the potential P effect on microbial communities was examined using various tools: for instance, some authors have used MiSeq or Hiseq amplicon sequencing from total DNA templates, RFLP analysis, DGGE fingerprints, or sequencing of targeted genes (i.e., alkaline phosphatase gene (*phoD*)) [70,91,105]. Others have assessed microbial communities by recording alkaline or phosphatase activities [80,106], by estimating microbial biomass [64,90] or quantifying phospholipid fatty acid (PLFAs) [80,90].

### 3.2. Occurrence of Microbial Communities with P-Dependent Structure and Composition

Phosphate inputs to soils are regularly reported to co-occur with changes in total microbial biomass, or functional profiles. For instance, Tang et al. [64] have shown significant increases in total microbial biomass-P inside the rhizosphere of intercrop plant species after P supplementation. By profiling bacterial and total PLFAs, Ali et al. [80] described changes in biochemical characteristics of microbial communities after P fertilization of cropping systems. Numerous studies also concluded that P can drive variation in microbial alpha and beta diversity indices across P fertilization treatment (RP, or TSP supplemented soils) in relation to the control (no P addition) [98] or according to the P level [94].

The identification of microbes with P-dependent abundances has also been reported. It is widely accepted that P is an important factor in the global biogeographic distribution of fungi [87,107,108]. Many studies have also concluded that P fertilization leads to changes in fungal communities, as developed for example by Wakelin et al. [86] in a long-term trial with P supplementation. Specifically, Yu et al. [31] observed that fungal community richness and species composition within roots varied with soil P contents. These authors also found that in axial roots β-diversity was similar at high or low P, whereas under low P conditions β-diversity was higher in lateral roots compared with high P. The species composition analysis they proposed, showed greater diversity for Basidiomycota at high-P and for Ascomycota at low-P in the lateral roots while under low-P conditions the diversity of Chytridiomycota was greater in the axial roots. In response to high or low soil P levels, changes in the composition of fungal communities have been observed in the bulk soil. Gomes et al. [94] also described significant differences in taxonomic composition linked to P level, mainly identified as variation in abundance of *Sordariomycetes*, *Dothideomycetes*, *Pleosporales* and *Phoma* across root or rhizosphere samples of maize.

Other studies also examined fertilizer induced changes on AMF, investigating mycorrhizal and non-mycorrhizal plant species [92,96]. Notably, high levels of mycorrhizal colonization under low P conditions and a lower hyphal and/or arbuscular mycorrhizal colonization rates of roots with high P level treatment have been repeatedly reported [105,109]. Accordingly, Gomes et al. [94] reported high abundance of *Glomeromycota* at low P content, with a prevalence of *Gigasporaceae*, *Scutellosporaceae* and *Racocetraceae*, in the root samples of maize. Similar to maize, Bodenhausen et al. [96] identified that the root microbiota of Petunia was enriched in *Glomeromycota* fungi in low-P conditions. However, in an apparent discrepancy, Silva et al. [98] reported that *Glomeromycota* were enriched in the RP-amended soils in comparison to other treatments (no P addition and TSP supplemented soil).

Wakelin et al. [89] concluded P-fertilizer application can either stimulate or depress AMF depending on the AMF species. They reported that the abundance of *Glomus intraradices* decreased as a component of the AMF community with P fertilization, while some taxa such as *G. mosseae*, *G. claroideum*, *Scutellospora*, and *Diversispora* respond positively to P fertilization and increased in abundance with the addition of fertilizer.

Similarly, Tang et al. [110] underlined the decrease of AMF colonization with P fertilization is not systematic in field-grown plant species, as P fertilization varied considerably with the plant species and indicator of root colonization by AMF. Thus, P-fertilization significantly reduced root colonization frequency and dramatically decreased arbuscular intensity under P treatments relative to the unfertilized treatment in durum wheat, but not in faba beans [110]

Similar to fungi, some data highlighted that the occurrence/abundance of bacterial taxa may vary according to P-dependent patterns. For example, Gomes et al. [30] found significant differences in the proportion of proteobacteria between P-rich and P-poor soil samples when looking for changes in bacterial taxa in the roots and rhizosphere of maize grown in oxisoils after fertilization with TSP. These significant differences were supported by the prevalence of γ-proteobacteria when P levels are high. They also reported a decrease in the Proteobacteria/Acidobacteria ratio in low-P conditions (without TSP addition) due to a positive effect of low-P content on Acidobacteria. Further analysis at the lower family level, illustrated fast-growing and copiotrophic *Enterobacteriaceae* and *Pseudomonadaceae* were enriched in high-P soils (with TSP), while roots from low P soils contained abundant slow-growing bacterial taxa, such as *Burkholderiaceae*. Comparing the effects of long-term fertilization practices with RP versus TPS on the microbiomes of maize in tropical oxisol soils, [98] pointed out decreases in the abundance of Proteobacteria with the TSP or RP amendments compared to the control. In addition, they observed a predominance of *Oxalobacteraceae* in the RP fertilized soil compared to the control and TSP treatments.

Trabelsi et al. [38] showed that the different bacterial groups that make up the bean rhizosphere had differential responses depending on the type of phosphate fertilizer used. Thus, following TSP inputs, Proteobacteria increased while Actinobacteria and Firmicutes decreased in the rhizosphere, whereas in the uncultivated soil, a decrease in Proteobacteria and an increase in Actinobacteria were observed. In contrast, the same authors observed stimulation of, mainly, Proteobacteria, Actinobacteria and Firmicutes when applying RP at the same rate in the uncultivated soil. However, increasing the RP rate induced a significant decrease in species richness concerning mainly Proteobacteria and Firmicutes in the uncultivated soil, while in the rhizosphere, the change in taxonomic structure was mainly due to an increase in Proteobacteria and a decrease in Actinobacteria.

Across a study comparing how the root-associated bacterial communities of mycorrhizal Petunia and the non-mycorrhizal Arabidopsis responded to being supplemented with fertilizer solutions that mainly varied in the P concentration, Bodenhausen et al. [96] established that the two plant species formed different microbial associations under low-P conditions. Notably, they found marked changes in the root microbiota of Arabidopsis between low- and high-P fertilization, while the opposite was reported by Robbins [92]. Bodenhausen et al. [96] also described that the root microbiota of Petunia was enriched in slow-growing bacterial taxa such as *Burkholderiaceae* in low-P conditions, while *Enterobacteriaceae* and *Pseudomonadaceae* were enriched in high-P conditions, which is in accordance to Gomes et al. [94] in maize.

As reported across setups based on inorganic fertilization, variation in P content following organic fertilizer application also significantly influenced the soil microbial community structure and composition. In particular, manure applications were favorable for different taxa; a part of the changes could be explained by soil C and N contents and C/N ratio [100,101].

### 3.3. Importance of Interkingdom Interactions among Plant-Associated Microbial Communities

Numerous biotic interactions occur inside the soil microbiota, where microbe-microbe interactions are important selective forces sculpting and likely stabilizing the complex microbial assemblages associated with plants [75,76]. Among these interactions, are interkingdom microbial associations between root fungi and the endobacteria they host in their cells [74]. Thus, numerous examples of bacterial endosymbionts have been reported; they mostly belong to the families *Burkholderiaceae* or related *Bacillaceae*, and live in intimate association with plant-associated fungi such as *Rhizophagus*, *Gigaspora*, *Laccaria*, *Mortierella*, *Mucoromycota*, *Ustilago*, *Rhizopus* sp. [111]. Lee et al. [112] proposed that the inter-kingdom interaction within the plant holobiont is governed by an interrelated temporal organization where arbuscular mycorrhizal symbiosis could be a determinant to coordinate circadian clocks in holobionts.

Moreover, the association between fungal hyphae and plant roots can lead to the establishment of a high interactive zone defined as the mycorrhizosphere that may stimulate other soil microorganisms, including PGPM. Hence, mycorrhizal plants, their colonizing fungi along with their mycorhizospheric bacteria and endobacteria, form a complex entity in the holobiont [113]. The evaluation of the effects of three sources of phosphorus in the presence or absence of AMF inoculation, on the microbial interactions of the soil led Gumiere et al. [39] to find that the percentage of P_2_O_5_ increased the number of bacteria-bacteria interactions while this percentage had the effect of reducing the number of fungi-fungi connections. They showed that AMF inoculation may drive a high percentage of variability inside the soil bacterial community, reaching 41% and they hypothesized that such a percentage may be associated with the recruitment of bacteria by AMF since the hypha-recruited bacteria may support the AMF for phosphorus acquisition.

In this sense, the analysis of microbial networks is now a privileged tool to assess microbial interactions in ecosystems. These analyses, pairwise comparisons between the abundance profiles of individual taxa, make it possible to identify positive, neutral or negative connections between the microbial partners of a plant. If such correlations do not necessarily predict causal relationships, they progressively increase the understanding of microbial communities. For example, co-occurrence network analysis has been used by Bodenhausen et al. [32] to characterize pairs or groups of microbes with similar abundance behavior along the P-fertilizer gradient. These authors did not find that P-rich conditions gave rise to groups of co-occurring OTUs (operational taxa units). On the other hand, they identified two major modules comprising groups of OTUs particularly abundant in P-poor conditions; the first module mainly includes Betaproteobacteria, Burkholderiales and Rhodocyclales [32]; the second module groups together a set of OTUs of the Glomeral order. An OTU assimilated to endobacteria closely associated with an AMF was revealed by their co-occurrence analysis. This OTU had an abundance consistent with the mycorrhizal OTUs identified along the P gradient [32]. The occurrence of these microbe-microbe interactions adds a further level of complexity to identifying and understanding how P influences soil microbial communities.

**Table 1 microorganisms-10-00609-t001:** Influence of P towards soil microbiome: Diversity of some selected experimental designs developed in microbial community profiling for assessing P-dependant shifts.

Experimental Design				Assessment of Shifts in Microbial Assemblages			References
Fertilization Practices P Sources	P Levels in Amendment	Ecosystem Or Culture Conditions	Plant Species Culture Duration before Plant Sampling	Soil Compartments	Targeted Microbial Communities	Diversity Analysis	
**Long-Term Fertilization Practices—Repetitive Inputs During the Long Term**							
~40 year fertilization trial Superphosphate Phosphate rock	188 SP; 250SP; 250RP; 376SP kg ha^−1^yr^−1^	pastoral agricultural system	grass/clover	soil	Actinobacteria Pseudomonas, AMF	Gene copie numbers (qPCR) PCR-DGGE profiles	[89]
~3 year fertilization trial Solutions of NaH2PO4	15 g m^−2^ year^−1^ in 2 monthly portions	tropical forests	tree species of a mixed forest	soil	Bacteria Fungi AMF	Microbial biomass Community composition (PLFA)	[90]
2 or 4 year fertilization trial Triple-super phosphate	10 g of P per m^2^·yr^−1^	Broad range of natural sites	native plants Growing season	Soil	Bacteria Archaea Fungi	Sequencing of gene markers; Alpha diversity; Taxonomic structure Functional gene composition	[97]
43 year fertilization trial Triple-super phosphate	no inputs 11 kg P ha^−1^ yr^−1^ 33 kg P ha^−1^ yr^−1^	field experiment	Intercropping *Faba bean*/ *Durum wheat* flowering stage	Rhizosphere Bulk soil	Actinobacteria, α-Proteobacteria Firmicutes	Microbial biomass Genes copies numbers (qPCR)	[110]
2 years trial NaH_2_PO_4_.2H_2_O.	5 g P m^−2^ yr^−1^ 15 g P m^−2^ yr^−1^ 30 g P m^−2^ yr^−1^	plantation	Subalpine spruce plantation	soil	Bacteria Fungi AMF	Microbial biomass Community composition (PLFA)	[104]
Fertilization since 1902 mineral fertilization (NPK) farmyard manure fertilization Combined farmyard manure and mineral fertilization	NPK = calcium ammonium nitrate+ superphosphate+ potassium chloride, 20t ha^−1^ of manure	field experiment	4-year crop rotation (*B. vulgaris*; *H. vulgare*; *S. tuberosum*; *T. aestivum*).	soil	Bacteria Fungi	microbial biomass Enzyme activities Sequencing of gene markers; Richness; Alpha diversity; Taxonomic structure	[100]
~30 years trial superphosphate Organic manure	40 kg P_2_O_5_ ha^−1^ year^−1^	field experiment	*Triticum aestivum* L.	soil	Bacteria Archea	Sequencing of gene markers;Alpha/beta diversity	[101]
3 years fertilization trial Triple superphosphate Crude rock phosphate	P2O adjusted: 4 kg/ha 100 kg/ha	field experiment	Maize 60 days	Root Rhizosphere	Bacteria Fungi AMF	Sequencing of gene markers; Alpha diversity; Taxonomic structure T-RFLP	[114]
Since 1949 Superphosphate Basic slag Alkali sinter phosphate 6-year crop rotation	0; 5 kg P ha^−1^ year^−1^	Greenhouse	*Arabidopsis thaliana* 7–8 weeks	Roots Rhizosphere Bulk soil	Bacteria Fungi	Sequencing of gene markers; Alpha/beta diversity; Taxonomic structure	[92]
long-term experiment	0; 150 kg ha^−1^	field experiment	Maize 10 weeks	Axial roots Lateral roots Bulk soil	Fungi	Sequencing of gene markers; Alpha/beta diversity; Taxonomic structure transcriptome sequencing	[115]
**Short term P fertilization/One-time phosphate fertilization/**							
Triple-super phosphate Rock phosphate	50 kg P ha^−1^ 50 kg P ha^−1^ 250 kg P ha^−1^	glasshouse Agricultural soil	*Phaseolus vulgaris* 10 weeks	Rhizosphere Bulk soil	Bacteria	PCR-TRFLP: Richness; Taxonomic structure	[102]
Potassium phosphate	0; 5; 10 and 20 kg P ha^−1^	greenhouse Agronomic soil	*Lolium perenne* 14 weeks	soil	Bacteria Fungi AMF	DGGE fingerprintings Sequencing of gene markers;Alpha/beta diversity Gene abundance(phoD) Phosphatase activity	[105]
Soils from low/High P area (4.4 mg/dm^3^ 5.3 mg/dm^3^) Additional superphosphate	90 kg/ha P_2_O_5_	field experiment	Maize 60 days	Roots Rhizosphere	Bacteria Fungi	Sequencing of gene markers; Alpha/beta diversity; Taxonomic structure	[94]
P-K or P-Na buffer	1; 20; 50 mM P	Phytochamber Agricultural soil	*Arabidopsis thaliana* 8 week	Roots Rhizosphere Bulk soil	Fungi	Sequencing of gene markers; Alpha diversity; Taxonomic structure Fungal Co-occurrence networks	[93]
Superphosphate Rock phosphate AMF inoculation (Rhizophagus clarus)	60 mg of P_2_O_5_ per kg	Greenhouse Agricultural soil	*Sugarcane* 120 days	Soil	Bacteria Fungi	DGGE analysis; Taxonomic structure Co-occurrence network	[103]
Nutritive solution KH2PO4	0; 0.03 mM; 1 mM; 5 mM	Climate chamber Agronomic soil+sand	*Petunia hybrida*/ *Arabidopsis thaliana* 10 weeks	Roots	AMF Fungal endobacteria	Sequencing of gene markers; Alpha diversity; Taxonomic structure Co-occurrence network	[96]

AMF: arbuscular mycorrhizal fungi. PLFA: phospholipid fatty acid analysis. TRFLP: Terminal-Restriction Fragments Length Polymorphisms. DGGE: Denaturing Gradient Gel Electrophoresis.

### 3.4. Identification of General Rules Explaining the Shifts in Microbiomes Following P Inputs Are Lacking

As detailed above, the literature documents numerous interactions between P and shifts in microbial community profiles. The data illustrate how P impacts microbial species compositions and diversity of bacterial and fungal communities in the bulk soil, the roots and in the rhizosphere, where different P-dependent taxa have been identified. Concomitantly, community responses have been captured at various taxonomic resolutions [92,93,94].

However, contradictory results have also been reported. For instance, Bodenhausen et al. [96] found marked root microbiota changes in *Arabidopsis* between low-P and high-P fertilized treatments, while Robbins et al. [92] did not. Silva et al. [98] obtained significantly higher Shannon and Simpson indices for bacteria in P added samples compared to control samples without P supplied. On the contrary, Silva and Nahas [114] and Toljander et al. [115] had originally described a higher bacterial diversity in the P unfertilized soil, while Huang et al. [104] indicated that soil microbes were insensitive to an elevated P availability in a subalpine spruce plantation.

Other research has looked more specifically at the possible impact of P on PSB. Ikoyi et al. [105] pointed out the relative abundance of bacterial genera, such as *Bacillus*, *Bradyrhizobium*, *Paenibacillus*, widely described elsewhere as genera harboring PSB strains, were significantly lower with P-fertilization compared to the control. In contrast, Gomes et al. [30] concluded that the relative abundance of taxa frequently associated with phosphate solubilization capabilities and more generally plant growth promotion did not necessarily reflect the effects of phosphate fertilization. Their findings were in line with those of Tang et al. [50] who concluded that the stimulation elicited by P fertilization was not limited to a specific group of PSB, but concerned the entire bacterial and fungal community. Similarly, at sites with extractable P levels ranging from 11.9 to 296.5 ppm, Fernandez et al. [59] did not observe any particular effect of P on PSB communities. Indeed, a review by Kour et al. [17] on bacterial taxa hosting PSB suggested that PSB are widely distributed in soils. Thus, these authors estimated that soil PSM accounts for 20–40% of the total population of the rhizosphere and about 10–15% of the total population of the bulk soil. They identified 551 P-solubilizers from the literature, and highlighted that PSM belongs to diverse phyla, most dominant being Proteobacteria (38%) and Firmicutes (22%), while only 1% of P solubilizers belonged to Bacteroidetes, and 6% to Actinobacteria. Among P-solubilizing fungi, 20% of P-solubilizing fungi were identified as Ascomycota; Basidiomycota represented 3%. Therefore, whether PSBs are impacted by P, strictly because of their PSM functionality, or rather, as components of an overall microbial community that is impacted as a whole, remains an open question. Further studies are needed to make a solid statement about the validity of the interaction.

Taken together, our review illustrates that large-scale conclusions are still hazardous and little consensus has emerged regarding possible key taxa with systematic dependence on P. Here, the diversity in experimental designs may explain the great complexity in analyzing the effect of P on the soil microbiome. Besides, studies of P impact on microbiomes typically focus on total or available P content data while changes in microbial community composition have been rarely analyzed and significantly correlated with the broad range of soil P compounds resulting from P cycling. Indeed, a soil matrix displays specific local environmental conditions with diverse biological, physical and chemical properties driving the dynamics of P compounds and the occurrence of a large diversity of P-compounds with different properties, in particular their bioavailability towards microbial nutrition and biomass. Consequently, we hypothesize the dynamics of P may have more significant effects on soil microbial communities than P supplementation *stricto sensu*. To support this hypothesis, in the next section, we revisit the literature on P speciation in soils, and some of the environmental or agronomic factors that influence the physicochemical equilibria of P compounds.

## 4. Assessing P-Impact on Microbial Communities to Identify Rules in P-Dependent Shifts, Require Appropriate Characterization of Amended P Forms and Their Fate

As reported above, a large diversity of P-compounds occurs in the soil. However, analysis of P impacts on soil microbiome generally focuses on total and available P, and there is a clear lack of in-depth analysis about the influence of the other P-forms linked to soil edaphic properties and P cycling. Yet, it would be relevant to consider that P dynamics may have more effects on soil microbial communities than P amendments stricto sensu. Accordingly, we must ask two questions. First, which range of the chemical P pool and species are precisely introduced as P-fertilizers in soils across the experimental designs? In unmanaged ecosystems, the P-content of soils can be predicted from the local soil properties resulting of transformations that occur on a geological time-scale and during soil development from the lithologic parent material [60,104]. Conversely, in agricultural systems, farming practices such as P fertilization, alter the P cycle by introducing different P compounds [63,116,117,118,119,120]. The mined RP is the raw material of manufactured chemical fertilizers. It can be profitably recovered from sedimentary inorgP deposits, igneous inorgP deposits and biogenic deposits. In addition, during the different industrial processes required to improve the purity of RP in P pentoxide (P_2_O_5_) [121], the tricalcium phosphate is converted diversely; consequently, fertilizers are available in over 100 different blends, notably with varying concentrations of nitrogen, phosphorus and potassium, sometimes also associated with varying amounts of heavy metals constituents as minor constituents in the ores, like arsenic, cadmium, chromium, lead, mercury, nickel, or vanadium [121]. As well, the P-forms in organic amendments [122,123,124] greatly vary depending, for instance, on animal species, age, diet and how manure has been stored [125,126]. In addition, organic sources used as P fertilizer usually result in higher soil organic matter content and in larger soil microbial biomass and activity, as well as higher soil organic C level, compared to soil receiving mineral fertilizers [39,79,117,127,128].

The second emerging question arising when studying impact of P inputs on microbial community, is: what is the fate of P-inputs shortly after application, according to soil properties? Indeed, a large diversity of P-compounds can be added to soils through P fertilization and many changes, more or less rapid and drastic, in the native and added P forms may be expected due to numerous drivers shaping the P-form diversity. For instance, a P fraction is immediately up taken by plant or immobilized by soil microbes. Furthermore, due to numerous physicochemical reactions that transformed the soluble to insoluble P pools in soils, up to 80% of P applied as fertilizer can be fixed into less or un-available fractions, shortly after application, mostly via sorption/precipitation with soil particles or microbial immobilization [123]. Due to such high P-fixing capacity of numerous soils, many agricultural soils have accumulated large amounts of P after past successive applications of chemical P fertilizer “in excess” of that required to support high yield of crop production. Consequently, to date, high legacy P stocks are available in some agricultural soils, but what could also be a supplementary P resource through full management of inherited soil P [129,130]. Furthermore, erosion or leaching can impact the introduced P-forms, leading consequently to modification of P-contents in sampled soils from different sites which are rarely analyzed when studying microbial community assemblages subsequently to fertilizer inputs. In addition, organic fertilizers can also help in mobilizing native P in soil through the action of organic acids and/or by chelation and sorption [101,131,132,133]. However, there is a lack of data to determine the extent to which P cycling may result in differences in soil microbial community profiles.

## 5. Perspectives

In this review, we discussed P soil status as a driver of microbial communities on one hand, and P-form diversity in soils on the other. We highlighted two constraints for the future in the field of plant microbiome research and for the formulation of microbial inoculants relevant for P plant nutrition [112]. Firstly, a deeper understanding of how multi-kingdom interactions inside the plant holobiont shape microbial communities is required. Here, meta-analysis to explore interactions, as well functional ecology approaches, and network analysis, may be relevant [76,130,131,132]. Second, we also suggest advancing concomitantly our knowledge about interactions between P and microbial communities in soils by taking into account the P dynamics and chemical form diversity and transformation. Here, we believe that in the future, multidisciplinary and integrated approaches should be preferred, in order to build quantitative models of P transformation in order to develop a dynamic approach to the phosphate fertility of soil. Thus, only joint advances in these different research areas developed in distinct laboratories will allow the design and deployment of effective microbial fertilizers efficient in contrasting conditions [84,133,134,135,136] with predictable behavior and robust results, in an efficient way at a large scale.

## Figures and Tables

**Figure 1 microorganisms-10-00609-f001:**
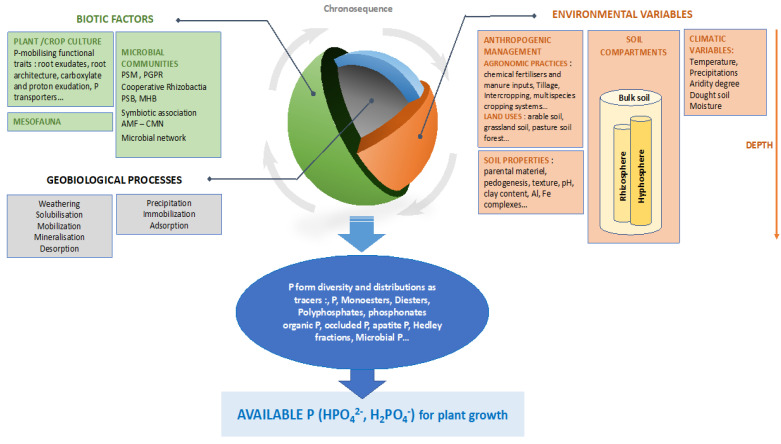
Conceptual representation of P dynamic drivers shaping P speciation. PSB: Phosphate solubilizing rhizobacteria: PSM: phosphate solubilizing microorganisms; MHB: mycorrhizal helper bacteria; CMN: common mycorrhizal network; PGPR: plant growth promoting rhizobacteria; AMF: arbuscular mycorrhizal fungi.

## Data Availability

Not applicable.

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
