# Peer review of "Diversity of Phosphate Chemical Forms in Soils and Their Contributions on Soil Microbial Community Structure Changes"

_microorganisms, 2022, doi:10.3390/microorganisms10030609_

Round 1

Reviewer 1 Report

The article “Soil phosphorus as a driver of microbial community profiles, and diversity of phosphate chemical forms in soils: a challenge for formulation of microbial P-solubilizing of bio fertilizers” compiled on a good theme and briefly discussed. However need improvement/ reframing of sentences at the several places for clear presentation:

Line-63- Delete   “without significant reduction of the crop yield”

Line 83-87 , communities is approached in different ways, each studying a different biotope, distinct 84 plant species, with various experimental designs including different agricultural management, repeated P inputs over time or not, and with different analytical tools for identifying  microbial community…….sentence is complex and hard understand, reframe the sentence

Line 135:  section 2.2. Reports of microbial communities with P-dependent structure and composition are available… looking odd

450-453 : reframe the sentence

455-456 Yang, et al. [101] developed maps of the global distribution of different P forms based on a global literature collection of profile data.. .. Seems incomplete, please add the outcome of this global distribution

Line 475-476 :  “of different P species during the evolution of a natural ecosystem can be controlled by changes in plant and microbial communities that use different forms of phosphorus differently”…. Elaborate different forms of phosphorus differently utilized by microbes

Author Response

The article “Soil phosphorus as a driver of microbial community profiles, and diversity of phosphate chemical forms in soils: a challenge for formulation of microbial P-solubilizing of bio fertilizers” compiled on a good theme and briefly discussed. However need improvement/ reframing of sentences at the several places for clear presentation:

Response: we are very grateful to the Reviewer for taking the time to think about this review seriously and provide us with the opportunity to present a more concise paper.

Line-63- Delete   “without significant reduction of the crop yield”

Response: this was deleted.

Line 83-87 , communities is approached in different ways, each studying a different biotope, distinct 84 plant species, with various experimental designs including different agricultural management, repeated P inputs over time or not, and with different analytical tools for identifying  microbial community…….sentence is complex and hard understand, reframe the sentence

Response: this sentence was rephrased as suggested by the Reviewer#1. However, its section was moved to page 8, L376-380, as per suggested by the Reviewer#2.

Line 135:  section 2.2. Reports of microbial communities with P-dependent structure and composition are available… looking odd

Response: we changed the tittle of sub-section 2.2.  as ‘Occurrence of microbial communities with P-dependent structure and composition’.

450-453 : reframe the sentence

Response: the sentence was rephrased.

Line 475-476 :  “of different P species during the evolution of a natural ecosystem can be controlled by changes in plant and microbial communities that use different forms of phosphorus differently”…. Elaborate different forms of phosphorus differently utilized by microbes

Response: we elaborated different forms of phosphorus differently utilized by microbes as fellow: 

“Notably, these drivers control accumulation of organic matter and modifications in P sources with occurrence of an increasing range of soil orgP during long term ecosystem development, relevant as substratum for further P-cycling and turnover by microbes”

Reviewer 2 Report

This review manuscript could help us understand the function and factors of soil phosphorus. This manuscript contained three aspects about soil phosphorus, the interactions/relationship between soil phosphorus and microbe had been well studied, the author should enrich this aspect at mechanism levels.I also suggested that author should firstly stated the this aspect “Diversity of phosphorus-forms and P-dynamic in soils” after the introduction section, and then discuss this aspect “Phosphorus as a driver of shifts in soil microbiomes”. Meanwhile, authors mainly list the published results, authors should summarize some similar papers to give the readers the common understanding in the revisions. The section 4 had a large range, while the authors could not give more detail information. Additionally, the title include “a challenge for formulation of microbial P-solubilizing of biofertilizers”, while the related information is not found in the subsequent content in this review manuscript.

Some detailed revisions should be made as followings: 

Title Could give us a brief title?

Lines 27-28 The phosphorus and P could be deleted in the second and third keywords.

Line 65 The full name of PGP is missing.

Lines 102-103 Define the different microbial associations, different plant tissues? and/or different microbial type?

Line 524 The author name is missing.

Line 633 Is the format of cited references right?

Table 1 there are many format mistakes in table 1, such as “m2” shoule be “m2” ...

Figure 1 This figure could not well summarize the content of this review.

Author Response

This review manuscript could help us understand the function and factors of soil phosphorus. This manuscript contained three aspects about soil phosphorus, the interactions/relationship between soil phosphorus and microbe had been well studied, the author should enrich this aspect at mechanism levels.I also suggested that author should firstly stated the this aspect “Diversity of phosphorus-forms and P-dynamic in soils” after the introduction section, and then discuss this aspect “Phosphorus as a driver of shifts in soil microbiomes”.

Response: we are very grateful to the reviewers for taking the time to think about this work seriously and provide us with helpful comments.

We changed the order of Sections 2 and 3 as suggested by the Reviewer and also updated reference numbers.

Meanwhile, authors mainly list the published results, authors should summarize some similar papers to give the readers the common understanding in the revisions.

Response: we considered the Reviewer’s suggestion and modified the text to address this comment.

The section 4 had a large range, while the authors could not give more detail information.

Response: we modified the text pf section 4 as per the Reviewer’s suggestion.

Additionally, the title include “a challenge for formulation of microbial P-solubilizing of biofertilizers”, while the related information is not found in the subsequent content in this review manuscript.

Response: we changed the title to reflect the content of this review. It is now: 'Diversity of phosphate chemical forms in soils and their contributions on soil microbial community structure changes'

Some detailed revisions should be made as followings: 

Title Could give us a brief title?

Response: we changed the title.

Lines 27-28 The phosphorus and P could be deleted in the second and third keywords.

Response: it has been corrected.

Line 65 The full name of PGP is missing.

Response: it was added.

Lines 102-103 Define the different microbial associations, different plant tissues? and/or different microbial type?

Response: it was modified as follow: 

“For example, it appears that non-mycorrhizal Arabidopsis thaliana and mycorrhizal Petunia sp. formed different microbial associations under the same low-P conditions [32] : notably, Burkholderiales (ex: Candidatus accumulibacter), Rhodocyclales (ex : Dechloromonas sp.) and Bdellovibrionales are the most abundant bacteria in Arabidopsis roots while only Burkholderiales (ex : Candidatus glomeribacter gigasporarum are associated in Petunia roots. Rhodocyclales as prevalent members.”

Line 524 The author name is missing.

Response: name was added.

Line 633 Is the format of cited references right?

Response:corrected.

Table 1 there are many format mistakes in table 1, such as “m2” shoule be “m2” ...

Response: corrected.

Figure 1 This figure could not well summarize the content of this review.

Response: the purpose of Figure 1 is to present the drivers that govern the speciation of P. We agree that this figure do not summarize this review article.